# Pore Structure Damages in Cement-Based Materials by Mercury Intrusion: A Non-Destructive Assessment by X-Ray Computed Tomography

**DOI:** 10.3390/ma12142220

**Published:** 2019-07-10

**Authors:** Xiaohu Wang, Yu Peng, Jiyang Wang, Qiang Zeng

**Affiliations:** College of Civil Engineering and Architecture, Zhejiang University, Hangzhou 310058, China

**Keywords:** non-destructive method, damage, mercury intrusion porosimetry, X-ray computed tomography

## Abstract

Mercury intrusion porosimetry (MIP) is questioned for possibly damaging the micro structure of cement-based materials (CBMs), but this theme still has a lack of quantitative evidence. By using X-ray computed tomography (XCT), this study reported an experimental investigation on probing the pore structure damages in paste and mortar samples after a standard MIP test. XCT scans were performed on the samples before and after mercury intrusion. Because of its very high mass attenuation coefficient, mercury can greatly enhance the contrast of XCT images, paving a path to probe the same pores with and without mercury fillings. The paste and mortar showed the different MIP pore size distributions but similar intrusion processes. A grey value inverse for the pores and material skeletons before and after MIP was found. With the features of excellent data reliability and robustness verified by a threshold analysis, the XCT results characterized the surface structure of voids, and diagnosed the pore structure damages in terms of pore volume and size of the paste and mortar samples. The findings of this study deepen the understandings in pore structure damages in CBMs by mercury intrusion, and provide methodological insights in the microstructure characterization of CBMs by XCT.

## 1. Introduction

Pore structure characteristics of cement-based materials (CBMs) importantly indicate their mechanical property and durability performance. Determining the pore structure of CBMs, however, still faces big challenges because (1) pore structure testing methods, more or less, have intrinsic shortages, and (2) the microstructure of cement hydrates is rather sensitive to environments [1]. Mercury intrusion porosimetry (MIP) is probably one of the most widely used techniques for characterizing the pore structure of CBMs due to its advantages of simple physic principle, broad pore rang (depending on the maximum pressure applied), and low costs in time and manpower (fast and easy operation and sample preparation). With those features, the pore structure characteristics by MIP may become a benchmark when assessing the pore structure of CBMs with different methods [2].

Because liquid mercury is hydrophobic to most solids, it cannot invade into pores spontaneously without sufficient external pressures. The surface forces of the mercury fronts in pores, inversely depending on the pore curvatures, will resist the forces applied. By recording the stepwise increased mercury volume with pressure, the volume–pressure data can be obtained. Those original data generally are not directly used without the specific relation between pressure and pore size. With the assumptions of cylindrical, size-graded and connected pores, the pore size-pressure relation gives: D=−4γcosθ/P, with *D*: the pore diameter; *P*: the applied pressure; γ: the surface tension of mercury; and θ: the contact angle between mercury and pore wall. This relation is known as the Washburn equation [3]. Through this simple equation, MIP provides various size-related pore parameters, such as accumulative and differential pore size distributions (PSDs), mean size, threshold size and fractal dimension [4,5,6,7].

Whilst MIP has been popularly used, its accuracy in pore structure characterization is always debatable [8,9,10,11,12,13,14]. Generally, the debates of MIP pore data focus on: (1) the microstructure damages by sample pretreatment (drying), (2) the oversimplifications of pore topology (geometry and connectivity), (3) the constants of MIP parameters, (4) the conformance effect of samples, and (5) the pore damages by high pressures during mercury intrusion. The first term on the microstructure damages by pretreatment is inevitable but may be mitigated by using the relatively mild drying methods (e.g., solvent exchange [1,4,15]). The second term on the oversimplifications of pore topology that are intrinsically related to the physical bases of MIP would yield the so-called “ink-bottle” effect [8]. To compensate for this shortage, a multi-cycled intrusion-extrusion test scheme was often applied [14,16,17]. The third term on the constant MIP parameters is argued with the significant influences of the contact angle between mercury fronts and pore walls [11,18]. The parameters of MIP may be greatly different when the tested pores are narrowed to nano sizes [18]. However, due to the lack of data at nano sizes, the improvements in size-associated MIP parameters for pore structure characterization are limited. The fourth term on the conformance effect of samples is rarely mentioned because the samples of CBMs generally have no big differences. Our recent tests, however, showed that the surface conformance control may greatly narrow the threshold pore size [13]. The last term on the pore structure damages of CBMs during mercury intrusion, albeit noticed by previous researcher [19,20,21], can hardly be quantitatively characterized.

Feldman [20] used a repeat-intrusion testing scheme to detect the possible pore structure alterations of blend cement pastes. Note that the repeat-intrusion testing scheme used by Feldman [20] is different from the multi-cycled intrusion-extrusion test that is often operated in a stepwise loading-unloading way without expelling the mercury entrapped in the pores. In his tests, the second mercury intrusion was operated after the entrapped mercury during the first intrusion was completely removed, so the differences in PSDs between the first and second MIP tests can reflect the sizes of the pores damaged. It was observed by Feldman that damages to the pore structure occurred at 70 MPa in the hydrated blends. Olson et al. [21] employed an environmental scanning electron microscopy (ESEM) to in situ observe the damages to pore structure of a hardened Portland cement paste by mercury intrusion. Analyses indicated that the connectivity of the pores between 1–10 μm was raised after the intrusion pressure reached the threshold value. However, the results of Feldman [20] may fail to capture the real damaged pore sizes due to the oversimplifications of pore topology by MIP as mentioned above. Despite of the direct and obvious evidence of pore structure damages by MIP documented by Olson et al. [21], the ESEM tests on the open samples without specific treatments would be unsafe to the operators because mercury is highly evaporable at room temperature and poison to humans. Therefore, seeking a non-destructive method to assess the microstructure damages of CBMs before and after MIP is urgently wanted. X-ray computed tomography (XCT) may be a preferable candidate because XCT not only is a non-destructive method, but also provides the component and spatial information of the object tested.

By delivering X-ray beams at different angles, numerous 2D radiographic projections of a scanned object can be gathered and treated in a digital geometry processing to construct the 3D digital structures of the object [22]. The continual methodological developments of XCT test make it extensively used for the phase characterization in CBMs for predicting their mechanical and transport performances [23,24,25,26]. Because mercury has much stronger X-ray absorptivity than any constituent in CBMs and other nonmetallic materials, the combination of XCT and MIP may provide an effective way to enhance the ability of XCT to detect the pores beyond the normal voxel resolution [27]. This further provides a routine to detect the pore damages of CBMs after MIP because mercury drops can be entrapped in the damaged and/or undamaged pores [13]. The method also generates significances for pore structure characterization because mercury entrapment may also, to some extent, reflect the connectivity of pores [28].

In the present study, XCT tests were operated on ordinary Portland cement (OPC) paste and mortar before and after mercury intrusion to evaluate their pore structure changes with deepened analyses and discussions. The findings of this study provide a new and effective routine to non-destructively characterize the pore structure damages of CBMs by MIP.

## 2. Materials and Experiments

### 2.1. Materials and Sample Preparation

A PI 42.5 OPC cement (corresponding to ASTM Type I) was used as the only binding phase to prepare the porous paste and mortar samples. The chemical component and physical properties of the cement are shown in Table 1. When preparing the samples, no agents were used to control the fluidity of the fresh paste and mortar slurries. However, to obtain the similar fluidity, the water-to-cement (*w*/*c*) ratios of 0.4 and 0.5 were adopted for the paste and mortar, respectively. Commercial standard quartz sands with the fineness modulus of 2.6 and the ${\mathrm{SiO}}_{2}$ content above 95% (Xiamen ISO Standard Sand Co., Ltd., Xiamen, China) were used as the fine aggregates to prepare the mortar. The cement/sand ratio was controlled as 1/3. Following standard casting, moulding and demoulding procedures, macro paste and mortar specimens were prepared, and then cured in a chamber with temperature at 20 ± 2 and relative humidity above 95%. After 28 days, the well cured specimens were crushed into small pieces (around 1 mL in volume or 2 g in mass) for further experiments.

The crushed OPC paste and mortar pieces were then immersed into pure ethanol to cease the hydration of cement. After two days of immersion in ethanol, the samples were then removed into an oven at 105 to expel the water physically absorbed in the pores. After 24 h, the samples were then stored in a sealed desiccator to eliminate the possible influences of water and carbon oxide in the air on the microstructure, and were readily prepared for XCT and MIP tests. Note that the drying temperature used here may be too severe to preserve the microstructure of the samples because the rapid water loss in the pores may yield high capillary stresses to damage the material matrix [1] and to alter the status of calcium-silicate-hydrate (C-S-H) gels [29]. However, because the microstructure alterations by drying are stable and will not recover during the following MIP and XCT tests, those alterations can be treated as the intrinsic microstructure features, and thus would be not considered here.

### 2.2. MIP Tests

The pre-dried samples were then placed into a sample chamber for MIP test in a device of Autopore IV 9510 (Micromeritics, Norcross, GA, USA). After a pre-equilibrium step to fill the gaps between the sample and chamber wall at 0.5 psi (3.45 KPa), pressures on mercury were automatically and stepwise raised to 60,000 psi (413.69 MPa) and then unloaded to certain values. With the equilibrium time of 10 s, an complete MIP test lasted about 130 min.

Immediately after the MIP tests finished, the samples were carefully and rapidly removed into small plastic tubes. A quick-hardening epoxy resin was rapidly poured into the tubes to completely cover the samples, and those tubes were quickly and tightly lidded (Figure 1). Those steps were to cease the leakages of mercury from the mercury-filled samples for diminishing the possible dangers to the technicians when handling those CBM samples.

### 2.3. XCT Tests

Before and after the MIP tests, XCT scans were performed on the samples by an X-CT scanner of Nikon XTH 255/320 LC (Nikon, Tokyo, Japan). For the XCT tests before and after MIP processes, the voltages to deliver the X-ray beams were set as 100 KeV and 150 KeV, respectively (Table 2; see below for detailed explanations). The penetrated X-ray beams were detected by a high-sensitive detector (DRZplus Scintillator with the pixels of 2000(h) × 2000(v)) at the back of the objects synchronously. During testing, the samples rotated in the rate of 12 /min. The collected data were then loaded into the software of VGStudio Max (version 3.1, Volume Graphics, Inc., Charlotte, NC, USA) for further analyses. Because the pre-MIP sample size was smaller than the post-MIP one (CBM sample plus tube), the pixel resolution of the former case was slightly higher (5.02 μm for the pre-MIP sample versus 5.60 μm for the post-MIP sample).

The reason for using different delivering voltages for the pre-and post-MIP tests was because mercury has far higher X-ray mass attenuation coefficient (MAC) than the main solid phases in the CBM samples. Figure 2 shows the MAC curves of mercury, ${\mathrm{SiO}}_{2}$, C-S-H and cement in the generally used photon energy interval (90–160 KeV). At 100 KeV (the photon energy used for the pre-MIP XCT tests), the MACs of ${\mathrm{SiO}}_{2}$, C-S-H and cement are rather close (0.17–0.21 $m^{2}/g$). The slight MAC differences among those phases will even become less at a higher photon energy (see the inserted panel in Figure 2), so to obtain the high quality images of the pre-MIP samples, the photon energy of 100 KeV was selected. When mercury was intruded into the CBM samples, the MACs are greatly altered. As displayed in Figure 2, the MAC of mercury is 25–30 times higher than that of ${\mathrm{SiO}}_{2}$, C-S-H and cement at 100 KeV. Such strong X-ray absorption by mercury would induce significant beam hardening artifacts [30]. Since the MAC gaps between mercury and the other phases will be narrowed with increasing photon energy (Figure 2), it is thus expected that a higher photon energy may bring less beam hardening artifacts. Our practices indeed indicated the images at the photon energy of 150 KeV would achieve high quality images for further analysis.

Because the samples used have irregular shapes and rough surfaces, it is unrealistic to analyse the entire volume of the samples. Instead, some volumes of interest (VOI) inside the samples (or region of interest in 2D analysis) were selected for image analysis, which can prevent edge effects and increase data efficiency. Due to the heterogeneity in microstructure and the chaos in pore structure of CBMs [6], a random selection of VOI that would be preferred for obtaining representative and reproductive results was not adopted here. In order to easily and precisely identify the same VOI of the samples before and after mercury intrusion, the microstructure of the VOI must have distinguished characteristics. In this study, we intentionally selected the VOIs containing big voids. Figure 3 shows an example of the best fit registration of two cubic VOIs in the pre-MIP paste sample. Clearly, with these big voids (dark circles in the VOIs shown in Figure 3), the same VOIs can be easily identified for the same sample after mercury intrusion.

## 3. Results and Discussion

### 3.1. MIP Outcomes

For pore structure characterization, the classic Washburn equation was used to interpret the MIP data with the mercury surface tension of 485 mN/m and the contact angle between mercury and substrate of 130. With those data, some characteristic pore parameters of the paste and mortar samples, i.e., total porosity, volume-median pore size, specific surface area and threshold pore size, can be evaluated (Table 3). Obviously, compared with mortar, paste showed the higher total porosity and specific surface area, the similar threshold pore size, but the lower volume-median pore size. The results are reasonably due to the fact that the impermeable sands in the mortar occupied more than 60% of the total volume. The looser compactness and more porous cement hydrates of the mortar induced by the higher *w*/*c* ratio, as well as the porous interfacial transition zones (ITZs) between cement matrix and aggregates, caused the higher volume-median pore size (independent of the absolute pore volumes), but remained unable to compensate for the reductions in porosity and specific surface area (Table 3). The similar threshold pore sizes between the paste and mortar suggested that the connected throats formed from the interparticle continuum had the similar widths.

Figure 4 shows the (top) accumulative and (bottom) differential PSDs of the paste and mortar. While the PSDs of the paste and mortar had the different shapes, they both displayed the similar five-stage characteristics: surface conformance, non-channel stage, capillaries by flaws and ITZs, capillaries by interparticle space, and gel pores.

Mercury first covered the open cracks, gaps, cavities and irregularities on the sample surfaces in relatively low pressures (termed as the surface conformance effect) [13,32]. Our previous study [13] suggested that the surface conformance effect might not be avoided because these cavities, cracks and flaws can be inevitably induced during sample pretreatments such as cutting and drying [1,15]. However, the volume increases at the very beginning stage of MIP by the surface conformance effect can be mitigated by controlling the exposed areas of the MIP samples [13]. In this study, the surface conformance effect was insignificant for both the paste and mortar samples (<0.005 mL/g).Later, almost no mercury intrusion was recorded between 2 μm and 100 μm (Figure 4). This meant that no open channels (not the pores inside the materials) in such size interval can be recognized by MIP, which was termed as the non-channel stage.As the size decreased further, the mercury increases of both the paste and mortar became obvious (Figure 4). Generally, for normally cured cement paste, these increases can be rarely observed [8]. In this study, the very severe drying scheme (105 ) was used, so the microstructure flaws or damages by drying [1] would account for the abnormal mercury rises in this stage. For the mortar sample, the porous ITZs, together with the capillary flaws by drying, were responsible for the higher PSD data (see the shadowed areas shown in Figure 4).After that, the intrusion volumes rose rapidly and significantly with obvious peaks around 70 nm (Figure 4). The peak size was identical to the threshold pore size form the percolated pore continuum [29,33,34]. Because of the `ink-bottle’ effect [8], the volumes at or below the threshold size could partially represent the capillaries of the interparticle space that remained unfilled by cement hydration. Compared with the mortar sample, the paste sample showed the faster raising rate and higher peak intensity because of the higher capillary pores.Under the higher pressures, the mercury rising rates became slower and the differential PSDs were depressed (Figure 4) because only limited space (mainly gel pores) was available to accommodate the mercury after the capillaries were filled. Since the MIP parameters in nano scales remained debatable [11,18], those data would not shed much light on gel pore characterization.

### 3.2. Threshold Analysis

Despite the fact that XCT is a powerful tool to non-destructively characterize the microstructure of various materials, the results, as cautioned elsewhere [22,27], are highly depending on the process of threshold segmentation. In this section, the effect of threshold segmentation on the reconstructed results of the pore phase was discussed, so the reliability and robustness of the damage diagnoses by XCT could be guaranteed.

Figure 5 shows the voxel-grey value distributions of a VOI in the paste sample before and after mercury intrusion. Generally, due to the lower X-ray absorption, the empty pores in a CBM sample were captured by the low grey value area, and the cement skeletons (including the hydration products and unhydrated cement clinkers) that can absorb more X-rays thus were represented by the high grey value area (Figure 5a). When mercury was intruded into the empty pores, the characteristic areas of grey value were exchanged. Specifically, the high grey value area represented the mercury-filled pores, while the low grey value area denoted the cement matrix (Figure 5b). This feature of grey-value inverse was recently used to determine the mercury drops entrapped in the pores of HCP samples with/without surface conformance control for pore structure characterization [13].

The voxel-grey value distributions displayed in Figure 5 clearly showed two peaks, so the threshold segmentation should be operated at the minimum between the two peaks. Here, to discuss the threshold sensitivity, three threshold points were selected, i.e., the middle, low (−5%) and high (+5%) threshold values shown in Figure 5a. In Figure 6, the 2D and 3D images of the pores segmented from the paste skeleton are comparatively plotted with the designed three threshold values. Apparently, no obvious differences can be seen from those images. To specifically compare the pore information in different scales, the volume-size plots of all the extracted pores in a VOI of the cement with three threshold values were illustrated in Figure 7. The results showed that both the number and size of those objects had no obvious differences (Figure 7). A similar threshold analysis was also performed on the post-MIP samples to obtain the appropriate threshold grey values with the reliability data.

Overall, the data of Figure 6 and Figure 7 implied that the XCT analyses used in this study provided reliable and robust pore structure information for identifying the pore structure damages of CBMs by MIP. In the following contexts, all discussions were based on the XCT results with the middle threshold segmentation.

### 3.3. Characteristics of XCT Results

Figure 8 and Figure 9 show the 2D and 3D representative images of a VOI and a localized big pore of the cement and mortar sample, respectively. In a much clearer way, the grey-value inverse of the pores before and after mercury intrusion can be displayed. For instance, the same pores were illustrated in the darkest color before MIP and the brightest color after MIP (Figure 9b,e).

Some features in microstructure characterization of CBMs by the combination of MIP and XCT can be pointed out from Figure 8 and Figure 9. Firstly, all the visible air voids in the paste and mortar samples were fully filled with mercury. For ordinary CBMs, the air voids with the size range of 10–500 μm and the content less then 2.5% were reported [8]. Those air voids, however, can not be detected by MIP because it only measures the open channels rather than the pore chambers [13]. This finding again made evident that MIP fails to detect the `ink-bottle’ like pores [8]. Secondly, the air voids in the paste and mortar samples showed different surface structures. Specifically, the surfaces of the voids in the paste were rather rough with and without mercury intrusion (Figure 8c,f), while those in the mortar were much smoother (Figure 9c,f). Although the akin rough pore surfaces of CBMs were documented [13,35], the mechanisms for the surface structure differences between paste and mortar remained unclear and deserved further rigorous studies. Thirdly, the mercury intrusion process enhanced the contrast of the mortar images to figure out the aggregates (quartz). As shown in Figure 9d, the aggregates were illustrated as the darkest phase (due to the lowest X-ray absorption, Figure 2) embedded in the much brighter cement matrix and the brightest mercury-filled voids. Because of the closely valued MACs of quartz, C-S-H and cement clinkers (Figure 2), separating quartz from the other two phases would be difficult. The post-MIP XCT test used in this study may provide an effective way to obtain the packing pattern of the aggregates in mortar.

Last but not least, the mercury intrusion paths were identified from the XCT images of the post-MIP mortar. For instance, Figure 10 displays a 2D XCT image of a local area of the mortar sample after mercury intrusion, where the bright areas along the ITZs around the aggregates indicated the thoroughly penetrated path to the air void. However, these mercury penetration paths cannot be identified from the XCT images of the post-MIP paste sample (Figure 8d) because the penetration sizes in the paste (0.2 μm [8]) would be beyond the resolution of our XCT tests.

### 3.4. Damage Diagnosis

We then used the XCT data of the pre-and post-MIP samples to diagnose whether or not the microstructure of those samples was damaged after MIP and in what sizes the damages occurred.

Figure 11 and Figure 12 comparatively plot the PSDs and the statistics in pore volume of VOIs, respectively, in the paste and mortar samples before and after mercury intrusion. Note that the minimum diameters of the reconstructed pores (around 10 μm) shown in Figure 11 and Figure 12 were higher than the minimum detectable pixel sizes (Table 2) because the voxel resolution for a 3D reconstructed object (depending on the geometry of the object) would be always lower than the 2D pixel resolution [22]. After the MIP tests finished, clearly, the total pore volume was increased from 0.29 ${\mathrm{mm}}^{3}$ to 0.31 ${\mathrm{mm}}^{3}$ by 6.7% for the paste sample (Figure 11a), and from 0.13 ${\mathrm{mm}}^{3}$ to 0.14 ${\mathrm{mm}}^{3}$ by 7.6% for the mortar sample (Figure 12a). In a statistic manner, the mean pore volume was largely augmented from $5\times 10^{-6}$
${\mathrm{mm}}^{3}$ to $8\times 10^{-5}$
${\mathrm{mm}}^{3}$ by around 16 times for the paste sample (Figure 11b), and slightly from $5\times 10^{-5}$
${\mathrm{mm}}^{3}$ to $6\times 10^{-5}$
${\mathrm{mm}}^{3}$ by 20% for the mortar sample (Figure 12b). However, the heavy increases in mean pore volume shown in Figure 11b may be misleading because the numbers of the pores recognized (especially the thin-sized pores) were largely decreased. The increases in pore volume and decreases in pore number for CBMs were in line with the results reported by Olson et al. [21] through ESEM observations. From Figure 11 and Figure 12, one could further read the sizes of the pores damaged directly. Two obvious pore volume increases shown in Figure 11a indicated that the damages mainly occurred to the pores of 100–200 μm and 300–500 μm for the paste sample. For the mortar sample, the damages concentrated at the size interval of 100–400 μm (Figure 12b).

### 3.5. Further Discussion

When mercury is enforced to invade into a porous CBM sample, the capillaries among the compacted particles and the porous cement hydrates deform to sustain the applied pressures. If the pressures are (even locally) higher than the strength of the phases in contact with the stressed mercury, damages take place. Those mechanically reasonable damages to CBMs by mercury intrusion can be schematically illustrated in Figure 13. Before MIP, the voids in a CBM sample (generally in 10–500 μm [8]) may be isolated by the material matrix consisting of the closely compacted cement particles and their hydration products (Figure 13a). The capillaries (channels) connecting those voids are generally too thin to be diagnosed by normal XCT. However, after mercury intrusion under sufficient pressures, the voids as well as these throats will be filled with mercury. Since mercury can strongly absorb the X-ray penetrated, the signals of the mercury-entrapped channels (albeit below the resolution of the XCT) may become detectable (Figure 13b). This is the regime applied in this study to diagnose the pore damages of CBM samples after mercury intrusion.

In our tests, the damaged pore sizes measured by XCT were much higher than the data obtained by Olson et al. [21], who found that the connectivity of the pores in the 1–10 μm size range was greatly increased, and the average size was enlarged from 1.60 μm to 2.36 μm after MIP. A much lower size of the damaged pores was reported by Feldman [20] (around 18 nm corresponding to the applied pressure of 70 MPa). The size differences between our data and those reported in the literature [20,21] were mainly due to the different methods used and different objects concerned. In Feldman’s tests [20], the pore damages were assessed by the PSD differences of cement blends before and after mercury intrusion, so the obtained sizes were always underestimated due to the intrinsic biases in the pore sizes of MIP, e.g., the ‘ink-bottle’ effect [8]. In the tests by Olson et al. [21], 2D images from limited local areas were obtained from ESEM. In our tests because of the limited resolution of the XCT used, the pores below 10 μm, mainly the throats to connect the voids [8], can not be detected. Instead, the damaged voids under high pressures were diagnosed.

While the present study reported obvious damages to the pore structure of the paste and mortar samples, several themes remained to be discussed further. Firstly, one must understand that the obvious enhancement in image contrast by mercury may induce some biases in pore structure characterization due to the beam hardening artifacts [27,36]. For example, if the voids were neighbours, the signals of X-ray beams may overlap so the individual pores as well as the connecting channels may be diagnosed as a big pore. This would significantly decrease the detected pore numbers but increase the pore volumes (Figure 13b). This regime may also partially account for the results shown in Figure 11 and Figure 12. Secondly, the pore structure alterations by the severe drying process (105 ) may induce additional variances when assessing the pore damages by mercury intrusion. It has been recognized that severe drying can greatly impact the packing patterns of C-S-H gels and the connectivity of pores [15,37]. Those may increase the difficulties in diagnosing the pore damages to CBMs after mercury intrusion. Future tests on the CBM samples with the milder drying schemes are preferred to mitigate this effect.

## 4. Conclusions

XCT is a powerful technique to non-destructively characterize the microstructure of CBMs. The significant differences in X-ray MACs between mercury and the phases in CBMs can greatly enhance the contrast gradients in XCT images and facilitate the reconstruction of 3D microstructure.MIP tests indicated that, compared with the mortar sample, the paste sample had the higher porosity and specific surface area, similar threshold pore size, but lower median pore size. The MIP PSDs of the paste and mortar samples showed the similarly five-stage intrusion curves but the different specific spectra. The drying at 105 brought additional flaws just before the threshold stage to the paste and mortar samples.The grey values for the pores and material skeletons in the CBM samples were inversely distributed due to the shifts in X-ray absorptivity when the pores were filled with mercury.A threshold analysis indicated that the obtained XCT results showed good reliability and robustness in pore phase segmentation.The surfaces of the voids in the paste were rough, while those in the paste were smooth. Mercury intrusion paths along the ITZs around aggregates in the mortar sample were visible in the post-MIP XCT images.Mercury intrusion in the paste and mortar samples caused the increases in pore volume and the decreases in pore number as determined by XCT. The results were consistent with those reported in the literature.

Overall, the damages to the pore structure of CBMs after mercury intrusion can be non-destructively diagnosed by XCT with quantitative parameters. Going beyond this, the combination of MIP and XCT may provide a powerful tool to probe the pore structure alterations in CBMs under different environments.

## Figures and Tables

**Figure 1 materials-12-02220-f001:**
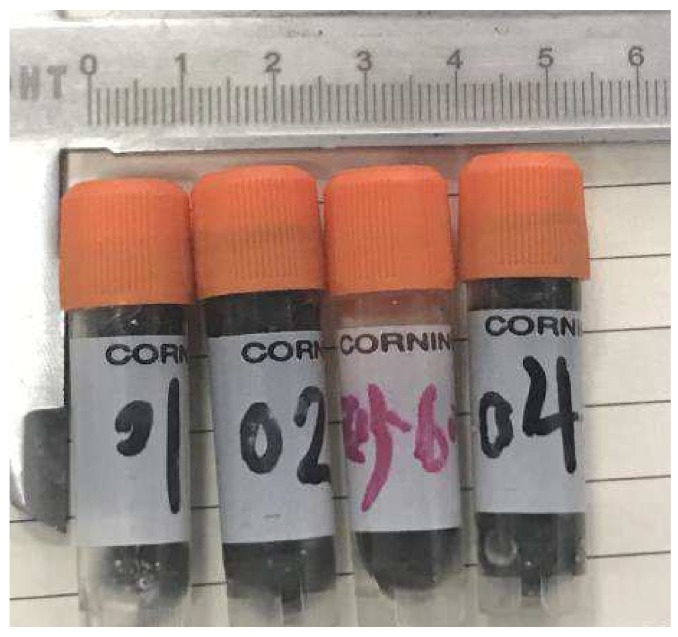
Paste and mortar samples encased in epoxy resin after mercury intrusion.

**Figure 2 materials-12-02220-f002:**
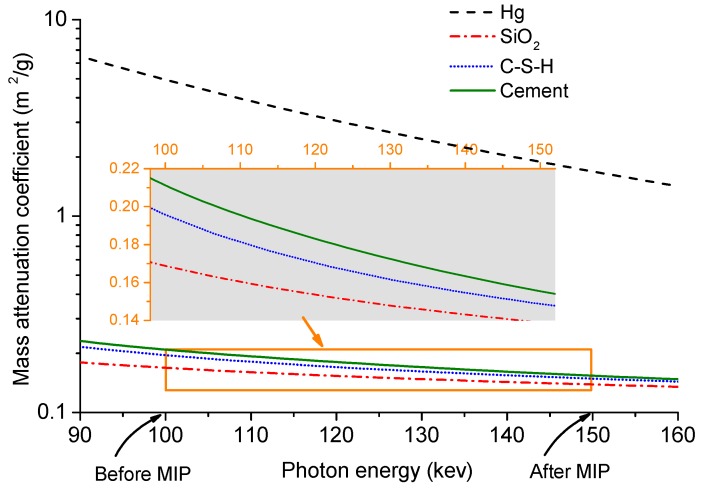
Spectra of mass attenuation coefficient of Hg, ${\mathrm{SiO}}_{2}$, C-S-H and cement between 90 KeV and 160 KeV (Data from Ref. [31] ).

**Figure 3 materials-12-02220-f003:**
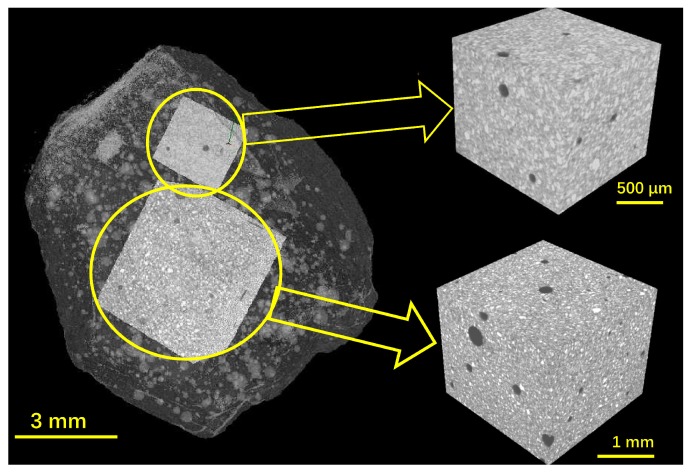
An example of VOI selections from a pre-MIP paste sample.

**Figure 4 materials-12-02220-f004:**
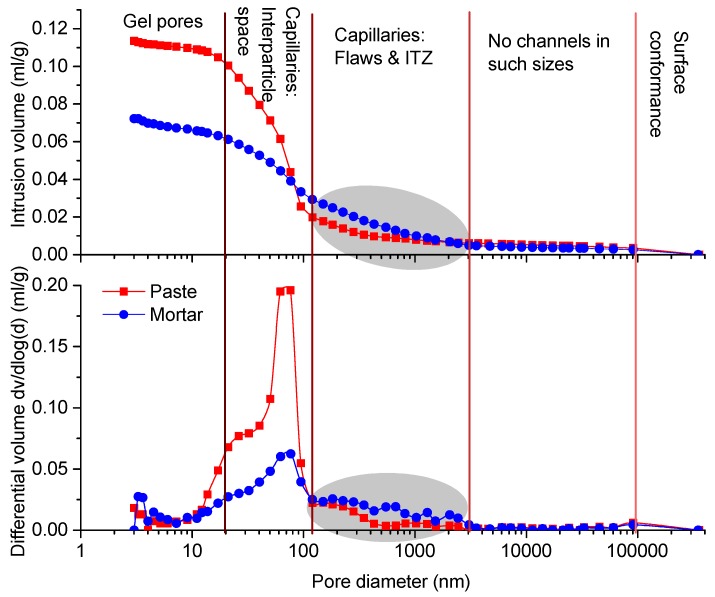
Accumulative (**top**) and differential (**bottom**) pore size distributions of paste and mortar samples (The specific contribution of ITZ in the mortar was singled out in the shadowed areas).

**Figure 5 materials-12-02220-f005:**
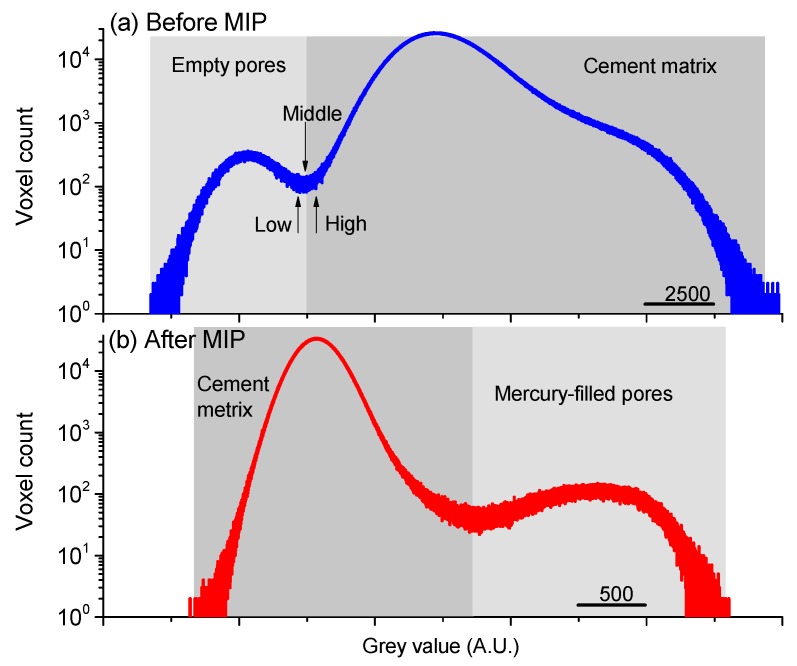
Voxel-grey value distributions of a VOI in the paste sample (**a**) before and (**b**) after mercury intrusion. Three threshold values (low, middle and high) were selected to test the influence of threshold process on pore segmentation.

**Figure 6 materials-12-02220-f006:**
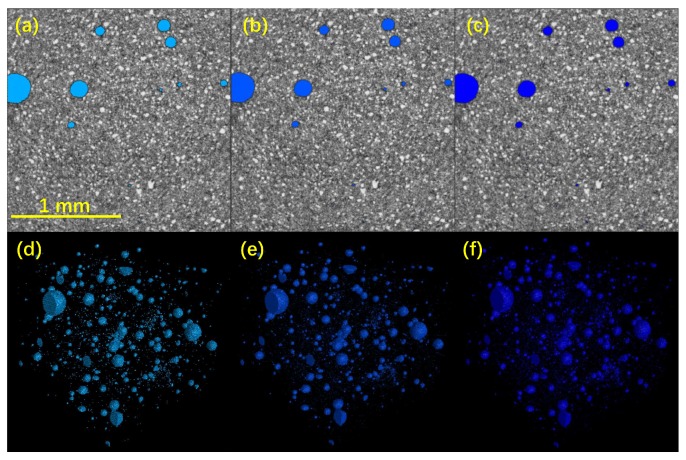
2D (**a**–**c**) and 3D (**d**–**f**) images of pores segmented from paste skeleton at (**a**,**c**) low, (**b**,**d**) middle, and (**c**,**f**) high threshold values from Figure 5.

**Figure 7 materials-12-02220-f007:**
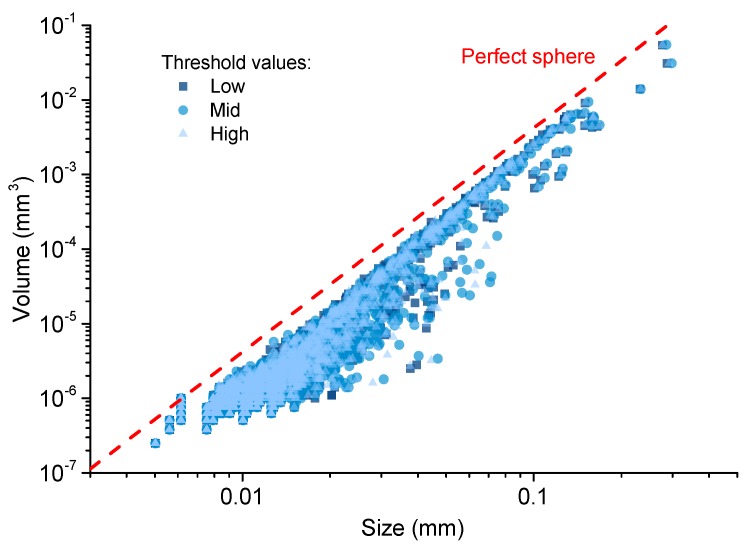
Volume-size distributions of the pores segmented from paste skeleton at low, middle, and high threshold values from Figure 5.

**Figure 8 materials-12-02220-f008:**
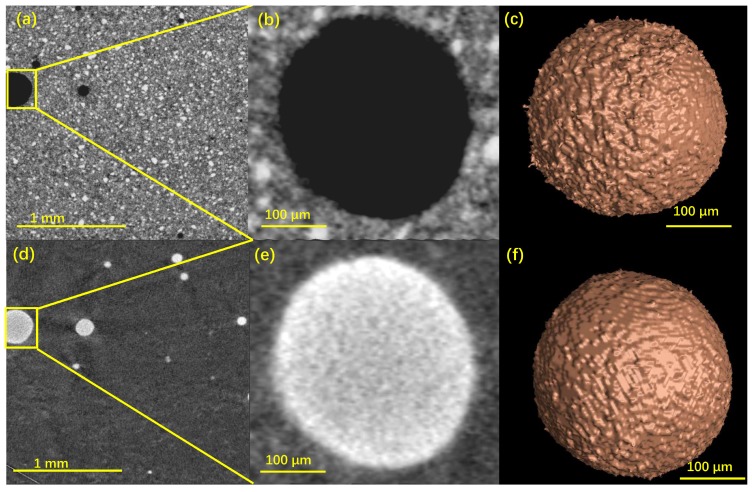
Representative images of a VOI in the cement sample (**a**–**c**) before and (**d**–**f**) after mercury intrusion: (**a**,**d**) the 2D sectional view with (**b**,**e**) the magnified pore and (**c**,**f**) its 3D structure.

**Figure 9 materials-12-02220-f009:**
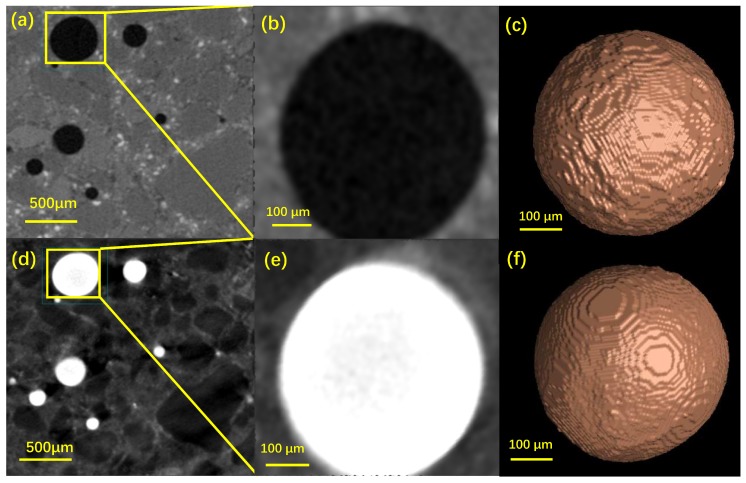
Representative images of a VOI in the mortar sample (**a**–**c**) before and (**d**–**f**) after mercury intrusion: (**a**,**d**) the 2D sectional view with (**b**,**e**) the magnified pore and (**c**,**f**) its 3D structure.

**Figure 10 materials-12-02220-f010:**
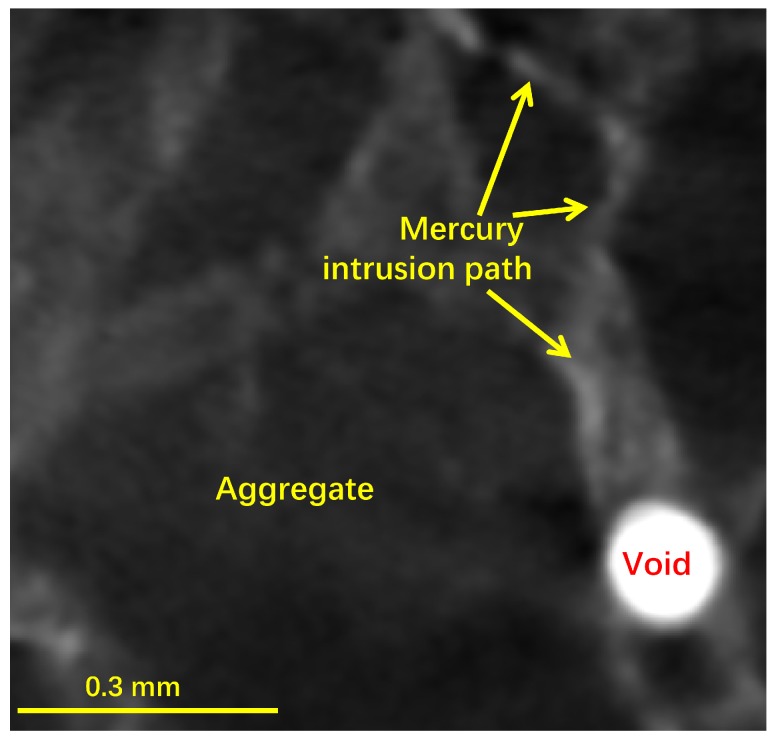
A 2D sectional image of a local area in the mortar showing a mercury intrusion path along the ITZs between aggregates and cement matrix to an air void.

**Figure 11 materials-12-02220-f011:**
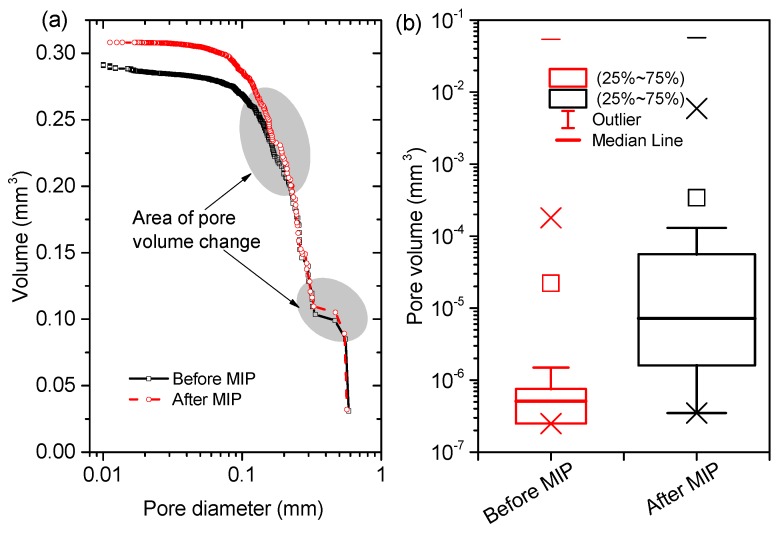
(**a**) comparative plots of pore size distribution of a VOI in the paste sample before and after mercury intrusion, and (**b**) the statistic results of the pore phase.

**Figure 12 materials-12-02220-f012:**
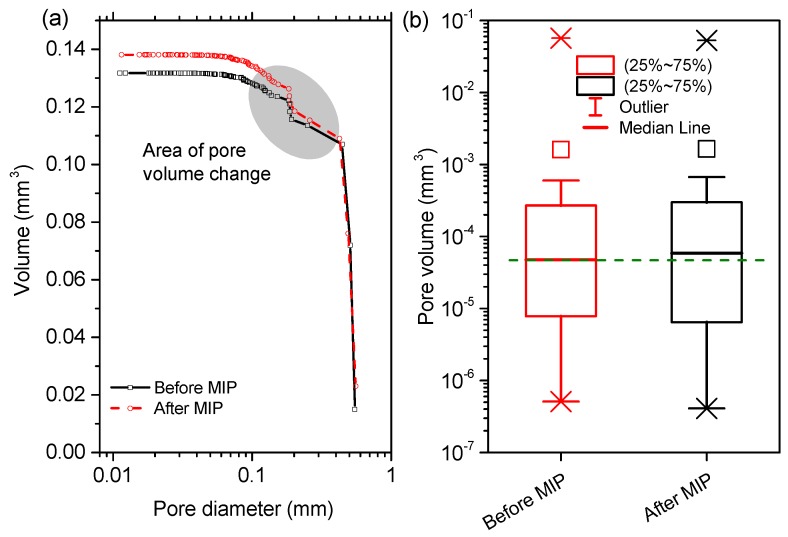
(**a**) comparative plots of pore size distribution of a VOI in the mortar sample before and after mercury intrusion, and (**b**) the statistic results of the pore phase.

**Figure 13 materials-12-02220-f013:**
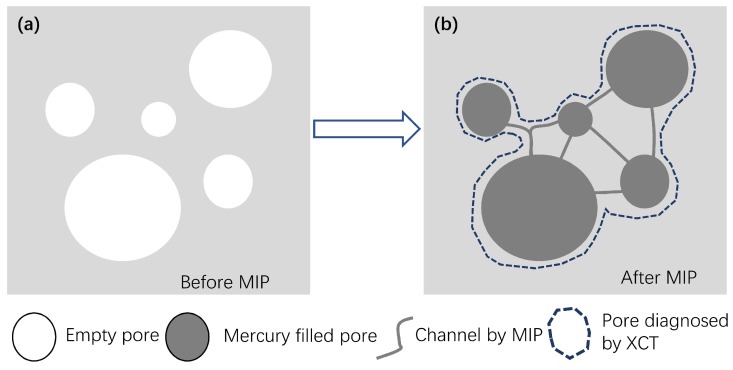
Mechanisms of pore damages induced by mercury intrusion under high pressures with the possibility of an overestimation on the pore size of a CBM sample after MIP (right) than that before MIP (left) by XCT.

**Table 1 materials-12-02220-t001:** Chemical component and physical properties of cement.

Oxides	Content (%)	Minerals	Content (%)	Physical Properties	Value
${\mathrm{SiO}}_{2}$	21.68	$C_{3}S$	57.34	Density (g/mL)	3.10
${\mathrm{Al}}_{2}O_{3}$	4.80	$C_{2}S$	18.09	Specific area (m${}^{2}$/kg)	345
${\mathrm{Fe}}_{2}O_{3}$	3.70	$C_{3}A$	6.47	Mean size (μm)	11
CaO	64.90	$C_{4}\mathrm{AF}$	11.25		
MgO	2.76	Others	6.04		
SO3	0.29				
${\mathrm{Na}}_{2}O\left(\mathrm{eq}\right)$	0.56				
CaO(f)	0.93				

**Table 2 materials-12-02220-t002:** XCT parameters used before and after MIP tests.

Condition	Voltage (KeV)	Pixel Resolution (μm)
Before MIP (Pre-MIP)	150	5.02
After MIP (Post-MIP)	100	5.60

**Table 3 materials-12-02220-t003:** Characteristic pore parameters of paste and mortar form MIP.

Sample	Total Porosity (%)	Volume-Median Pore Size (nm)	Specific Surface Area ($m^{2}/g$)	Threshold Pore Size (nm)
Paste	20.0	66.1	12.3	76.5
Mortar	15.1	85.1	9.3	76.9

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
