# Peer review of "Pore Structure Damages in Cement-Based Materials by Mercury Intrusion: A Non-Destructive Assessment by X-Ray Computed Tomography"

_materials, 2019, doi:10.3390/ma12142220_

Round 1

Reviewer 1 Report

Generally, an interesting draft in order to evaluate the possible errors and destructive damages in the MIP experiments. The techniques which were utilized are acceptable. Some minor editing might be needed (for instance, Figure 11 and 12 b not d).

-Is the difference between XCT results before and after MIP is just due to destructive damages of MIP? or it can be partially contributed to XCT errors?

_if the MIP has a strong destructive effect, why not repeating MIP cycles and compare the XCT results after each repeat to see if the difference is increasing or not? I think this way you can provide stronger evidence. 

Author Response

Response to reviewer 1

Generally, an interesting draft in order to evaluate the possible errors and destructive damages in the MIP experiments. The techniques which were utilized are acceptable. Some minor editing might be needed (for instance, Figure 11 and 12 b not d).

[Answer & Modification:] We thank the reviewer for the insightful comments to improve the quality of this study.

We corrected our mistakes of the figure captions (see the revised captions of Figures 11 and 12).

-Is the difference between XCT results before and after MIP is just due to destructive damages of MIP? or it can be partially contributed to XCT errors?

[Answer & Modification:] We thank the reviewer for the insightful comments to improve the quality of this study.

We also noticed the possible errors in pore structure characterization by XCT: a, the beam hardening artifacts due to the very strong X-ray absorption by mercury; b, the pore segmentation by an inappropriate threshold operation. In order to mitigate these errors, we have taken several processes

1, To decrease the possible beam hardening artifacts, we promoted the X-ray delivering energy from 100 KeV to 150 KeV to decrease the mass attenuation coefficient of Hg. This factor was discussed in lines 146-147, page 5.

2, To verify the reliability of pore segmentation in the post-MIP samples, a threshold analysis was performed (Figure R1, Please find the figure in the attached response letter). The results indicate that except for the case of 6% over threshold, all the other cases show the higher porosity for the post-MIP samples. This means that even with some variances in the threshold operation, the porosity of the post-MIP samples is always larger than that of the pre-MIP ones, which indeed evidences the pore damages of the paste and mortar sample induced by MIP. This factor was discussed in lines 229-231, page 9

_if the MIP has a strong destructive effect, why not repeating MIP cycles and compare the XCT results after each repeat to see if the difference is increasing or not? I think this way you can provide stronger evidence.

[Answer:] We thank the reviewer for the insightful comments to improve the quality of this study.

Indeed, Feldman (Ref. 20) used the repeating MIP test to verify whether or not mercury intrusion will damage the pore structure of cement-based materials. But his testing scheme is different from the generally used quick repeating MIP test. Instead, the samples after MIP require a very long time heating (more than 360 hours at 35 Celsius degree) to remove the mercury entrapped in pores. The very long time heating will change the microstructure of the samples. Furthermore, since mercury is poison to human, this repeating MIP test would not be a preferable method to determine the pore structure damage. In our test, the post-MIP samples were sealed in epoxy resin, so any further treatment on the samples will be safe. This issue was discussed in lines 122-124, page 4.

Reviewer 2 Report

The manuscript is original and well-written. The conclusions are supported by relevant discussion and findings. Minor remark prior to publication:

-Conclusions should appear in text form. Note that the phrase "An XCT-MIP-XCT testing scheme was designed to probe the pore structure damages of a paste and a mortar" should be not considered as conclusion

Author Response

Response to reviewer 2

The manuscript is original and well-written. The conclusions are supported by relevant discussion and findings. Minor remark prior to publication:

-Conclusions should appear in text form. Note that the phrase "An XCT-MIP-XCT testing scheme was designed to probe the pore structure damages of a paste and a mortar" should be not considered as conclusion

[Answer:] Agree and thank the reviewer for the comments to improve the quality of this study.

[Modification:] 1, We deleted this item;

2, We modified an item of our conclusion as "Mercury intrusion in the paste and mortar samples caused the increases in pore volume and the decreases in pore number as determined by XCT." (lines 337-338, page 14).

Round 2

Reviewer 1 Report

Thank you for the proper response. The current version is acceptable.